# Self-Imitation Learning via Generalized Lower Bound Q-learning

**Yunhao Tang**
Columbia University
yt2541@columbia.edu

## Abstract

Self-imitation learning motivated by lower-bound Q-learning is a novel and effective approach for off-policy learning. In this work, we propose a n-step lower bound which generalizes the original return-based lower-bound Q-learning, and introduce a new family of self-imitation learning algorithms. To provide a formal motivation for the potential performance gains provided by self-imitation learning, we show that n-step lower bound Q-learning achieves a trade-off between fixed point bias and contraction rate, drawing close connections to the popular uncorrected n-step Q-learning. We finally show that n-step lower bound Q-learning is a more robust alternative to return-based self-imitation learning and uncorrected n-step, over a wide range of continuous control benchmark tasks. The implementation is available at https://github.com/robintyh1/nstep-sil.

## 1 Introduction

Learning with off-policy data is of central importance to scalable reinforcement learning (RL). The traditional framework of off-policy learning is based on importance sampling (IS): for example, in policy evaluation, given trajectories $(x_t, a_t, r_t)_{t=0}^{\infty}$ generated under behavior policy $\mu$, the objective is to evaluate Q-function $Q^{\pi}(x_0, a_0)$ of a target policy $\pi$. Naive IS estimator involves products of the form $\pi(a_t \mid x_t)/\mu(a_t \mid x_t)$ and is infeasible in practice due to high variance. To control the variance, a line of prior work has focused on operator-based estimation to avoid full IS products, which reduces the estimation procedure into repeated iterations of off-policy evaluation operators [1–3]. Each iteration of the operator requires only local IS ratios, which greatly stabilizes the update.

More formally, such operators $\mathcal{T}$ are designed such that their fixed points are the target Q-function $\mathcal{T}Q^{\pi} = Q^{\pi}$. As such, these operators are *unbiased* and conducive to theoretical analysis. However, a large number of prior work has observed that certain *biased* operators tend to have significant empirical advantages [4–6]. One notable example is the uncorrected $n$-step operator, which directly bootstraps from $n$-step target trajectories without IS corrections [4]. The removal of all IS ratios biases the estimate, but allows the learning signal to be propagated over a longer horizon (in Section 2, we will characterize such effects as contraction rates). Indeed, when behavior trajectories are unlikely under the current policy, small IS ratios $\pi(a_t \mid x_t)/\mu(a_t \mid x_t)$ quickly cut off the learning signal. In general, there is a trade-off between the fixed point bias and contraction rates. Empirical findings suggest that it might be desirable to introduce bias in exchange for faster contractions in practice [7].

Recently, self-imitation learning (SIL) has been developed as a family of novel off-policy algorithms which facilitate efficient learning from highly off-policy data [8–10]. In its original form, SIL is motivated as lower bound Q-learning [11]. In particular, let $Q_L(x, a) \leq Q^{\pi^*}(x, a)$ denote a lower bound of the optimal Q-function $Q^{\pi^*}$. Optimizing auxiliary losses which encourage $Q_{\theta}(x, a) \geq Q_L(x, a)$ could significantly speed up learning with the trained Q-function $Q_{\theta}(x, a)$. Such auxiliary losses could be extended to actor-critic algorithms with stochastic policies [8]: SIL suggests optimizing a policy $\pi_{\theta}(a \mid x)$ by maximizing an objective similar to $[Q^{\mu}(a \mid x) - V^{\pi_{\theta}}(x)]_+ \log \pi_{\theta}(a \mid x)$,

where $V^{\pi_\theta}(x)$ is the value-function for policy $\pi_\theta$, with $[x]_+ := \max(0, x)$. The update is intuitively reasonable: if a certain actions $a$ is high-performing under behavior policy $\mu$, such that $Q^\mu(x, a) > V^{\pi_\theta}(x)$, the policy $\pi_\theta(a \mid x)$ should imitate such actions.

On a high-level, SIL is similar to the uncorrected $n$-step update in several aspects. With no explicit IS ratios, both methods entail that off-policy learning signals propagate over long horizons without being *cut-off*. As a result, both methods are biased due to the absence of proper corrections, and could be seen as trading-off fixed point bias for fast contractions.

**Main idea.** In this paper, we make several theoretical and empirical contributions.

- **Generalized SIL**. In Section 3, we propose generalized SIL which strictly extends the original SIL formulation [8]. Generalized SIL provides additional flexibility and advantages over the original SIL: it learns from partial trajectories and bootstraps with learned Q-function; it applies to both stochastic and deterministic actor-critic algorithms.
- **Trade-offs.** In Section 4, we formalize the trade-offs of SIL. We show that generalized SIL trades-off contraction rates with fixed point bias in a similar way to uncorrected $n$-step [7]. Unlike uncorrected $n$-step, for which fixed point bias could be either positive or negative, the operator for SIL induces *positive* bias, which fits the motivation of SIL to move towards optimal Q-functions.
- **Empirical.** In Section 5, we show generalized SIL outperforms alternative baseline algorithms.

## 2 Background

Consider the standard formulation of markov decision process (MDP). At a discrete time $t \geq 0$, an agent is in state $x_t \in \mathcal{X}$, takes action $a_t \in \mathcal{A}$, receives a reward $r_t = r(x_t, a_t) \in \mathbb{R}$ and transitions to a next state $x_{t+1} \sim p(\cdot \mid x_t, a_t) \in \mathcal{X}$. A policy $\pi(a \mid x) : \mathcal{X} \mapsto \mathcal{P}(\mathcal{A})$ defines a map from state to distributions over actions. The standard objective of RL is to maximize the expected cumulative discounted returns $J(\pi) := \mathbb{E}_\pi[\sum_{t \geq 0} \gamma^t r_t]$ with a discount factor $\gamma \in (0, 1)$.

Let $Q^\pi(x, a)$ denote the Q-function under policy $\pi$ and $Q^\pi \in \mathbb{R}^{|\mathcal{X}| \times |\mathcal{A}|}$ its vector form. Denote the Bellman operator as $\mathcal{T}^\pi$ and optimality operator as $\mathcal{T}^*$ [12]. Let $\pi^*$ be the optimal policy, i.e. $\pi^* = \arg\max_\pi J(\pi)$. It follows that $Q^\pi, Q^{\pi^*}$ are the unique fixed points of $\mathcal{T}^\pi, \mathcal{T}^*$ respectively [13]. Popular RL algorithms are primarily motivated by the fixed point properties of the Q-functions (or value functions): in general, given a parameterized Q-function $Q_\theta(x, a)$, the algorithms proceed by minimizing an empirical Bellman error loss $\min_\theta \mathbb{E}_{(x,a)}[(Q_\theta(x, a) - \mathcal{T}Q_\theta(x, a))^2]$ with operator $\mathcal{T}$. Algorithms differ in the distribution over sampled $(x, a)$ and the operator $\mathcal{T}$. For example, Q-learning sets the operator $\mathcal{T} = \mathcal{T}^*$ for value iteration and the samples $(x, a)$ come from an experience replay buffer [14]; Actor-critic algorithms set the operator $\mathcal{T} = \mathcal{T}^{\pi_\theta}$ for policy iteration and iteratively update the policy $\pi_\theta$ for improvement, the data $(x, a)$ could be either on-policy or off-policy [15–18].

### 2.1 Elements of trade-offs in Off-policy Reinforcement Learning

Here we introduce elements essential to characterizing the trade-offs of generic operators $\mathcal{T}$ in off-policy RL. For a complete review, please see [7]. Take off-policy evaluation as an example: the data are generated under a behavior policy $\mu$ while the target is to evaluate $Q^\pi$. Consider a generic operator $\mathcal{T}$ and assume that it has fixed point $\tilde{Q}$. Define the contraction rate of the operator as $\Gamma(\mathcal{T}) := \sup_{Q_1 \neq Q_2} \| \mathcal{T}(Q_1 - Q_2) \|_\infty / \| Q_1 - Q_2 \|_\infty$. Intuitively, operators with small contraction rate should have *fast* contractions to the fixed point. In practical algorithms, the quantity $\mathcal{T}Q(x, a)$ is approximated via stochastic estimations, denoted as $\tilde{\mathcal{T}}Q(x, a)$. All the above allows us to define the bias and variance of an operator $\mathbb{B}(\mathcal{T}) := \| \tilde{Q} - Q^\pi \|_2^2, \mathbb{V}(\mathcal{T}) := \mathbb{E}_\mu[\| \tilde{\mathcal{T}}Q - \mathcal{T}Q \|_2^2]$ evaluated at a Q-function $Q$. Note that all these quantities depend on the underlying MDP $M$, though when the context is clear we omit the notation dependency.

Ideally, we seek an operator $\mathcal{T}$ with small bias, small variance and small contraction rate. However, it follows that these three aspects could not be optimized simultaneously for a general class of MDPs $M \in \mathcal{M}$

$$\sup_{M \in \mathcal{M}} \{\mathbb{B}(\mathcal{T}) + \sqrt{\mathbb{V}(\mathcal{T})} + \frac{2r_{\max}}{1 - \gamma}\Gamma(\mathcal{T})\} \geq I(\mathcal{M}), \tag{1}$$

where $r_{\max} := \max_{x,a} r(x,a)$ and $I(\mathcal{M})$ is a information-theoretic lower bound [7]. This inequality characterizes the fundamental trade-offs of these three quantities in off-policy learning. Importantly, we note that though the variance $\mathbb{V}(\mathcal{T})$ is part of the trade-off, it is often not a major focus of algorithmic designs [4, 7]. We speculate it is partly because in practice the variance could be reduced via e.g. large training batch sizes, while the bias and contraction rates do not improve with similar techniques. As a result, henceforth we focus on the trade-off between the bias and contraction rate.

## 2.2 Trading off bias and contraction rate

Off-policy operators with unbiased fixed point $\mathbb{B}(\mathcal{T}) = 0$ are usually more conducive to theoretical analysis [3, 7]. For example, Retrace operators $\mathcal{R}_c^{\pi,\mu}$ are a family of off-policy evaluation operators indexed by trace coefficients $c(x,a)$. When $c(x,a) \leq \pi(a \mid x)/\mu(a \mid x)$, these operators are unbiased in that $\mathcal{R}_c^{\pi,\mu} Q^\pi = Q^\pi$, resulting in $\mathbb{B}(\mathcal{R}_c^{\pi,\mu}) = 0$. One popular choice is $c(x,a) = \min\{\bar{c}, \pi(a \mid x)/\mu(a \mid x)\}$ such that the operator also controls variance $\mathbb{V}(\mathcal{R}_c^{\pi,\mu})$ [3] with $\bar{c}$.

However, many prior empirical results suggest that bias is not a major bottleneck in practice. For example, uncorrected $n$-step update is a popular technique which greatly improves DQN [14] where the RL agent applies the operator $\mathcal{T}_{\text{nstep}}^{\pi,\mu} := (\mathcal{T}^\mu)^{n-1} \mathcal{T}^\pi$ where $\pi, \mu$ are target and behavior policies respectively [5, 6]. Note that since $\mathcal{T}_{\text{nstep}}^{\pi,\mu} Q^\pi \neq Q^\pi$, the $n$-step operator is biased $\mathbb{B}(\mathcal{T}_{\text{nstep}}^{\pi,\mu}) > 0$ [7]. However, its contraction rate is small due to uncorrected updates $\Gamma(\mathcal{T}_{\text{nstep}}^{\pi,\mu}) \leq \gamma^n$. On the other hand, though Retrace operators have unbiased fixed point, its contraction rates are typically high due to small IS, which *cut off* the signals early and fail to bootstrap with long horizons. The relative importance of contraction rate over bias is confirmed through the empirical observations that $n$-step often performs significantly better than Retrace in challenging domains [6, 7]. Such observations also motivate trading off bias and contraction rates in an adaptive way [7].

## 2.3 Self-imitation Learning

**Maximum entropy RL.** SIL is established under the framework of maximum-entropy RL [19–23], where the reward is augmented by an entropy term $r_{\text{ent}}(x,a) := r(x,a) + c\mathcal{H}^\pi(x)$ and $\mathcal{H}^\pi(x)$ is the entropy of policy $\pi$ at state $x$, weighted by a constant $c > 0$. Accordingly, the Q-function is $Q_{\text{ent}}^\pi(x_0, a_0) := \mathbb{E}_\pi[r_0 + \sum_{t \geq 1}^\infty \gamma^t (r_t + c\mathcal{H}^\pi(x_t))]$. The maximum-entropy RL objective is $J_{\text{ent}}(\pi) := \mathbb{E}_\pi[\sum_{t \geq 0}^\infty \gamma^t (r_t + c\mathcal{H}^\pi(x_t))]$. Similar to standard RL, we denote the optimal policy $\pi_{\text{ent}}^* = \arg\max_\pi J_{\text{ent}}(\pi)$ and its Q-function $Q_{\text{ent}}^{\pi_{\text{ent}}^*}(x,a)$.

**Lower bound Q-learning.** Lower bound Q-learning is motivated by the following inequality [8],

$$Q_{\text{ent}}^{\pi_{\text{ent}}^*}(x,a) \geq Q_{\text{ent}}^\mu(x,a) = \mathbb{E}_\mu[r_0 + \sum_{t \geq 1}^\infty \gamma^t (r_t + c\mathcal{H}^\mu(x_t))], \tag{2}$$

where $\mu$ is an arbitrary behavior policy. Lower bound Q-learning optimizes the following objective with the parameterized Q-function $Q_\theta(x,a)$,

$$\min_\theta \mathbb{E}_\mathcal{D}[([Q^\mu(x,a) - Q_\theta(x,a)]_+)^2], \tag{3}$$

where $[x]_+ := \max(x, 0)$. The intuition of Eqn.(3) is that the Q-function $Q_\theta(x,a)$ obtains learning signals from all trajectories such that $Q_{\text{ent}}^\mu(x,a) > Q_\theta(x,a) \approx Q^{\pi_\theta}(x,a)$, i.e. trajectories which perform better than the current policy $\pi_\theta$. In practice $Q_{\text{ent}}^\mu(x,a)$ could be estimated via a single trajectory $Q_{\text{ent}}^\mu(x,a) \approx \tilde{R}^\mu(x,a) := r_0 + \sum_{t \geq 1}^\infty \gamma^t (r_t + c\mathcal{H}^\mu(x_t))$. Though in Eqn.(3) one could plug in $\hat{R}^\mu(x,a)$ in place of $Q^\mu(x,a)$ [8, 10], this introduces bias due to the double-sample issue [24], especially when $\hat{R}^\mu(x,a)$ has high variance either due to the dynamics or a stochastic policy.

**SIL with stochastic actor-critic.** SIL further focuses on actor-critic algorithms where the Q-function is parameterized by a value-function and a stochastic policy $Q_\theta(x,a) := V_\theta(x) + c\log \pi_\theta(a \mid x)$. Taking gradients of the loss in Eqn.(3) with respect to $\theta$ yields the following loss function of the value-function and policy. The full SIL loss is $L_{\text{sil}}(\theta) = L_{\text{value}}(\theta) + L_{\text{policy}}(\theta)$.

$$L_{\text{value}}(\theta) = \frac{1}{2}([\hat{R}^\mu(x,a) - V_\theta(x)]_+)^2, \quad L_{\text{policy}}(\theta) = -\log \pi_\theta(a \mid x)[\tilde{R}^\mu(x,a) - V_\theta(x)]_+. \tag{4}$$

# 3 Generalized Self-Imitation Learning

## 3.1 Generalized Lower Bounds for Optimal Q-functions

To generalize the formulation of SIL, we seek to provide generalized lower bounds for the optimal Q-functions. Practical lower bounds should possess several desiderata: **(P.1)** they could be estimated using off-policy partial trajectories; **(P.2)** they could bootstrap from learned Q-functions.

In standard actor-critic algorithms, partial trajectories are generated via behavior policy $\mu$ (for example, see [25, 18, 26]), and the algorithm maintains an estimate of Q-functions for the current policy $\pi$. The following theorem states a general lower bound for the max-entropy optimal Q-function $Q_{ent}^{\pi_{ent}^*}$. Additional results on generalized lower bounds of the optimal value function $V^{\pi^*}$ could be similarly derived, and we leave its details in Theorem 3 in Appendix C.

**Theorem 1.** *(proof in Appendix A) Let $\pi_{ent}^*$ be the optimal policy and $Q_{ent}^{\pi_{ent}^*}$ its Q-function under maximum entropy RL formulation. Given a partial trajectory $(x_t, a_t)_{t=0}^n$, the following inequality holds for any $n$,*

$$Q_{ent}^{\pi_{ent}^*}(x_0, a_0) \geq L_{ent}^{\pi,\mu,n}(x_0, a_0) := \mathbb{E}_\mu[r_0 + \gamma c \mathcal{H}^\mu(x_1) + \sum_{t=1}^{n-1} \gamma^t(r_t + c\mathcal{H}^\mu(x_{t+1})) + \gamma^n Q_{ent}^\pi(x_n, a_n)]$$

(5)

By letting $c = 0$, we derive a generalized lower bound for the standard optimal Q-function $Q^{\pi^*}$

**Lemma 1.** *Let $\pi^*$ be the optimal policy and $Q^{\pi^*}$ its Q-function under standard RL. Given a partial trajectory $(x_t, a_t)_{t=0}^n$, the following inequality holds for any $n$,*

$$Q^{\pi^*}(x_0, a_0) \geq L^{\pi,\mu,n}(x_0, a_0) := \mathbb{E}_\mu[\sum_{t=0}^{n-1} \gamma^t r_t + \gamma^n Q^\pi(x_n, a_n)].$$

(6)

When $\pi = \pi^*$, Lemma 1 reduces to the lower bounds applied in [11]. We see that the $n$-step lower bounds $L_{ent}^{\pi,\mu,n}$ satisfy both desiderata **(P.1)(P.2)**: $L_{ent}^{\pi,\mu,n}$ could be estimated on a single trajectory and bootstraps from learned Q-function $Q_\theta(x, a) \approx Q^\pi(x, a)$. When $n \to \infty$, $L_{ent}^{\pi,\mu,n} \to Q^\mu$ and we arrive at the lower bound employed by the original SIL [8]. The original SIL does not satisfy **(P.1)(P.2)**: the estimate of $Q^\mu$ requires full trajectories from finished episodes and does not bootstrap from learned Q-functions. In addition, because the lower bound $L_{ent}^{\pi,\mu}(x, a)$ bootstraps Q-functions at a finite step $n$, we expect it to partially mitigate the double-sample bias of $\hat{R}^\mu(x, a)$. Also, as the policy $\pi$ improves over time, the Q-function $Q^\pi(x, a)$ increases and the bound $L^{\pi,\mu,n}$ improves as well. On the contrary, the standard SIL does not enjoy such advantages.

## 3.2 Generalized Self-Imitation Learning

**Generalized SIL with stochastic actor-critic.** We describe the generalized SIL for actor-critic algorithms. As developed in Section 2.3, such algorithms maintain a parameterized *stochastic* policy $\pi_\theta(a \mid x)$ and value-function $V_\theta(x)$. Let $\hat{L}_{ent}^{\pi,\mu,n}(x, a)$ denote the sample estimate of the $n$-step lower bound, the loss functions are

$$L_{value}^{(n)}(\theta) = \frac{1}{2}([\hat{L}_{ent}^{\pi,\mu,n}(x, a) - V_\theta(x)]_+)^2, L_{policy}^{(n)}(\theta) = -\log \pi_\theta(a \mid x)[\hat{L}_{ent}^{\pi,\mu,n}(x, a) - V_\theta(x)]_+.$$

(7)

Note that the loss functions in Eqn.(7) introduce updates very similar to A2C [25]. Indeed, when removing the threshold function $[x]_+$ and setting the data distribution to be on-policy $\mu = \pi$, we recover the $n$-step A2C objective.

**Generalized SIL with deterministic actor-critic.** For continuous control, temporal difference (TD)-learning and deterministic policy gradients have proven highly sample efficient and high-performing [15, 27, 23]. By construction, the generalized $n$-step lower bounds $L_{ent}^{\pi,\mu,n}$ adopts $n$-step

TD-learning and should naturally benefit the aforementioned algorithms. Such algorithms maintain a parameterized Q-function $Q_\theta(x, a)$, which could be directly updated via the following loss

$$L_{\text{qvalue}}^{(n)}(\theta) = \frac{1}{2}([\hat{L}_{\text{ent}}^{\pi,\mu,n}(x, a) - Q_\theta(x, a)]_+)^2. \tag{8}$$

Interestingly, note that the above update Eqn.(8) is similar to $n$-step Q-learning update [4, 5] up to the threshold function $[x]_+$. In Section 4, we will discuss their formal connections in details.

**Prioritized experience replay.** Prior work on prioritized experience replay [28, 29] proposed to sample tuples $(x_t, a_t, r_t)$ from replay buffer $\mathcal{D}$ with probability proportional to Bellman errors. We provide a straightforward extension by sampling proportional to the lower bound loss $[\hat{L}_{\text{ent}}^{\pi,\mu,n}(x, a) - Q_\theta(x, a)]_+$. This reduces to the sampling scheme in SIL [8] when letting $n \to \infty$.

# 4 Trade-offs with Lower Bound Q-learning

When applying SIL in practice, its induced loss functions are optimized jointly with the base loss functions [8]: in the case of stochastic actor-critic, the full loss function is $L(\theta) := L_{\text{ac}}(\theta) + L_{\text{sil}}(\theta)$, where $L_{\text{ac}}(\theta)$ is the original actor-critic loss function [25]. The parameter is then updated via the gradient descent step $\theta = \theta - \nabla_\theta L(\theta)$. This makes it difficult to analyze the behavior of SIL beyond the plain motivation of Q-function lower bounds. Though a comprehensive analysis of SIL might be elusive due to its empirical nature, we formalize the lower bound arguments via RL operators and draw connections with $n$-step Q-learning. Below, we present results for standard RL.

## 4.1 Operators for Generalized Lower Bound Q-learning

First, we formalize the mathematical operator of SIL. Let $Q \in \mathbb{R}^{|\mathcal{X}| \times |\mathcal{A}|}$ be a vector-valued Q-function. Given some behavior policy $\mu$, define the operator $\mathcal{T}_{\text{sil}}Q(x, a) := Q(x, a) + [Q^\mu(x, a) - Q(x, a)]_+$ where $[x]_+ := \max(x, 0)$. This operator captures the defining feature of the practical lower bound Q-learning [8], where the Q-function $Q(x, a)$ receives learning signals only when $Q^\mu(x, a) > Q(x, a)$. For generalized SIL, we similarly define $\mathcal{T}_{\text{n,sil}}Q(x, a) := Q(x, a) + [(\mathcal{T}^\mu)^{n-1}\mathcal{T}^\pi Q(x, a) - Q(x, a)]_+$, where $Q(x, a)$ is updated when $(\mathcal{T}^\mu)^{n-1}\mathcal{T}^\pi Q(x, a) > Q(x, a)$ as suggested in Eqn.(7,8).

In practice, lower bound Q-learning is applied alongside other main iterative algorithms. Henceforth, we focus on policy iteration algorithms with the Bellman operator $\mathcal{T}^\pi$ along with its $n$-step variant $(\mathcal{T}^\mu)^{n-1}\mathcal{T}^\pi$. Though practical deep RL implementations adopt additive loss functions, for theoretical analysis we consider a convex combination of these three operators, with coefficients $\alpha, \beta \in [0, 1]$.

$$\mathcal{T}_{\text{n,sil}}^{\alpha,\beta} := (1 - \beta)\mathcal{T}^\pi + (1 - \alpha)\beta\mathcal{T}_{\text{n,sil}} + \alpha\beta(\mathcal{T}^\mu)^{n-1}\mathcal{T}^\pi \tag{9}$$

## 4.2 Properties of the operators

**Theorem 2.** *(proof in Appendix B) Let $\pi, \mu$ be target and behavior policy respectively. Then the following results hold:*

- ***Contraction rate.*** $\Gamma(\mathcal{T}_{n,sil}^{\alpha,\beta}) \leq (1 - \beta)\gamma + (1 - \alpha)\beta + \alpha\beta\gamma^n$. *The operator is always contractive for $\alpha \in [0, 1], \beta \in [0, 1)$. When $\alpha > \frac{1-\gamma}{1-\gamma^n}$, we have for any $\beta \in (0, 1)$, $\Gamma(\mathcal{T}_{n,sil}^{\alpha,\beta}) \leq \gamma' < \gamma$ for some $\gamma'$.*

- ***Fixed point bias.*** $\mathcal{T}_{n,sil}^{\alpha,\beta}$ *has a unique fixed point $\tilde{Q}^{\alpha,\beta}$ for any $\alpha \in [0, 1], \beta \in [0, 1)$ such that $(1 - \alpha)\beta < 1$. This fixed point satisfies the bounds $Q^{\eta\pi+(1-\eta)\mu^{n-1}\pi} \leq \tilde{Q}^{\alpha,\beta} \leq Q^{\pi^*}$, where $Q^{\eta\pi+(1-\eta)\mu^{n-1}\pi}$ is the unique fixed point of operator $\eta\mathcal{T}^\pi + (1 - \eta)\mathcal{T}^{\mu^{n-1}\pi}$ with $\eta = \frac{1-\beta}{1-\beta+\alpha\beta}$.*

To highlight the connections between uncorrected $n$-step and SIL, we discuss two special cases.

- When $\alpha = 1$, $\mathcal{T}_{\text{n,sil}}^{\alpha,\beta}$ removes all the lower bound components and reduces to $(1 - \beta)\mathcal{T}^\pi + \beta(\mathcal{T}^\mu)^{n-1}\mathcal{T}^\pi$. This recovers the trade-off results discussed in [7]: when $\beta = 1$, the operator becomes uncorrected $n$-step updates with the smallest possible contraction rate $\Gamma(\mathcal{T}_{\text{n,sil}}^{\alpha,\beta}) \leq \gamma^n$, but the fixed point $\tilde{Q}^{\alpha,\beta}$ is biased. In general, there is no lower bound on the fixed point so that its value could be arbitrary depending on both $\pi$ and $\mu$.

- When $\alpha \in (\frac{1-\gamma}{1-\gamma^n}, 1]$, $\mathcal{T}_{n,sil}^{\alpha,\beta}$ combines the lower bound operator. Importantly, unlike uncorrected $n$-step, now the fixed point is lower bounded $\tilde{Q}^{\alpha,\beta} \geq Q^{\eta\pi+(1-\eta)\mu^{n-1}\pi}$. Because such a fixed point bias is lower bounded, we call it *positive bias*. Adjusting $\alpha$ creates a trade-off between contraction rates and the positive fixed point bias. In addition, the fixed point bias is safe in that it is upper bounded by the optimal Q-function, $\tilde{Q}^{\alpha,\beta} \leq Q^{\pi^*}$, which might be a desirable property in cases where over-estimation bias hurts the practical performance [30, 27]. In Section 5, we will see that such positive fixed point bias is beneficial to empirical performance, as similarly observed in [11, 8, 10]. Though $\mathcal{T}_{n,sil}^{\alpha,\beta}$ does not contract as fast as the uncorrected $n$-step operator $(\mathcal{T}^\mu)^{n-1}\mathcal{T}^\pi$, it still achieves a bound on contraction rates strictly smaller than $\mathcal{T}^\pi$. As such, generalized SIL also enjoys fast contractions relative to the baseline algorithm.

**Empirical evaluation of Q-function bias.** To validate the statements made in Theorem 2 on the bias of Q-functions, we test with TD3 for a an empirical evaluation [27]. At a given time in training, the bias at a pair $(x, a)$ is calculated as the difference between Q-function network prediction and an unbiased Monte-Carlo estimate of Q-function for the current policy $\pi$, i.e. $Q_\theta(x, a) - \hat{Q}^\pi(x, a)$[1]. Figure 1 shows the mean $\pm 0.5$std of such bias over time, with mean and std computed over visited state-action pairs under $\pi$. In general, the bias of TD3 is small, which is compatible to observations made in [27]. The bias of TD3 with uncorrected $n = 5$-step spreads over a wider range near zero, indicating significant non-zero bias on both sides. For TD3 with generalized SIL $n = 5$, the bias is also spread out but the mean bias is significantly greater than zero. This implies that SIL generally induces a positive bias in the fixed point. In summary, these observations confirm that neural network based Q-functions $Q_\theta(x, a)$ display similar biases introduced by the corresponding exact operators.

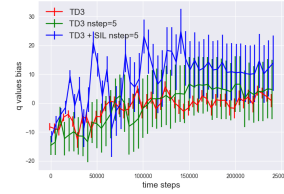

Figure 1: Bias of Q-function networks with twin-delayed deep deterministic policy gradient (TD3) variants on the WalkerStand task.

# 5 Experiments

We seek to address the following questions in the experiments: **(1)** Does generalized SIL entail performance gains on both deterministic and stochastic actor-critic algorithms? **(2)** How do the design choices (e.g. hyper-parameters, prioritized replay) of generalized SIL impact its performance?

**Benchmark tasks.** For benchmark tasks, we focus on state-based continuous control. In order to assess the strengths of different algorithmic variants, we consider similar tasks *Walker*, *Cheetah* and *Ant* with different simulation backends from OpenAI gym [31], DeepMind Control Suite [32] and Bullet Physics Engine [33]. These backends differ in many aspects, e.g. dimensions of observation and action space, transition dynamics and reward functions. With such a wide range of varieties, we seek to validate algorithmic gains with sufficient robustness to varying domains. There are a total of 8 distinct simulated control tasks, with details in Appendix D.

## 5.1 Deterministic actor-critic

**Baselines.** We choose TD3 [27] as the baseline algorithm which employs a deterministic actor $\pi_\phi(x)$. TD3 builds on deep deterministic policy gradient (DDPG) [15] and alleviates the over-estimation bias in DDPG via delayed updates and double critics similar to double Q-learning [34, 30]. Through a comparison of DDPG and TD3 combined with generalized SIL, we will see that over-estimation bias makes the advantages through lower bound Q-learning much less significant. To incorporate generalized SIL into TD3, we adopt an additive loss function: let $L_{TD3^{(n)}}(\theta)$ be the $n$-step TD3 loss function and $L_{sil}^{(m)}(\theta)$ be the $m$-step generalized SIL loss. The full loss is $L(\theta) :=$

$L_{\text{TD3}}^{(n)}(\theta) + \eta L_{\text{sil}}^{(m)}(\theta)$ with some $\eta \geq 0$. We will use this general loss template to describe algorithmic variants for comparison below.

**Return-based SIL for TD3.**    A straightforward extension of SIL [8] and optimality tightening [11] to deterministic actor-critic algorithms, is to estimate the return $\hat{R}^\mu(x_t, a_t) := \sum_{t' \geq t} \gamma^{t-t'} r_{t'}$ on a single trajectory $(x_t, a_t, r_t)_{t=0}^\infty$ and minimize the lower bound objective $([\hat{R}^\mu(x,a) - Q_\theta(x,a)]_+)^2$. Note that since both the policy and the transition is deterministic (for benchmarks listed above), the one-sample estimate of returns is exact in that $\hat{R}^\mu(x,a) \equiv R^\mu(x,a) \equiv Q^\mu(x,a)$. In this case, return-based SIL is exactly equivalent to generalized SIL with $n \to \infty$.

**Evaluations.**    We provide evaluations on a few standard benchmark tasks in Figure 2 as well as their variants with delayed rewards. To facilitate the credit assignment of the training performance to various components of the generalized SIL, we compare with a few algorithmic variants: 1-step TD3 ($n = 1, \eta = 0$); 5-step TD3 ($n = 5, \eta = 0$); TD3 with 5-step generalized SIL ($n = 1, \eta = 0.1, m = 5$); TD3 with return-based SIL ($n = 1, \eta = 0.1, m = \infty$). Importantly, note that the weighting coefficient is fixed $\eta = 0.1$ for all cases of generalized SIL. The training results of selected algorithms are shown in Figure 2. We show the final performance of all baselines in Table 1 in Appendix D.

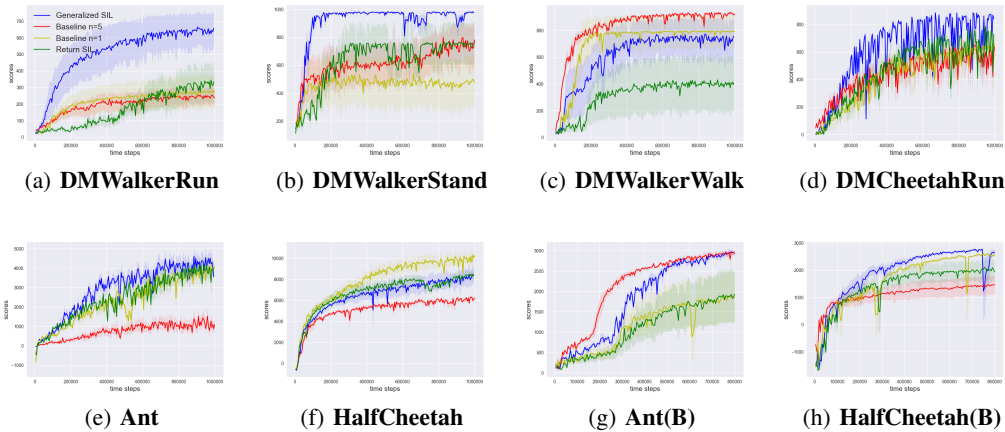

(a) **DMWalkerRun**  (b) **DMWalkerStand**  (c) **DMWalkerWalk**  (d) **DMCheetahRun**

(e) **Ant**  (f) **HalfCheetah**  (g) **Ant(B)**  (h) **HalfCheetah(B)**

Figure 2: Standard evaluations on 8 benchmark tasks. Different colors represent different algorithmic variants. Each curve shows the mean $\pm 0.5$std of evaluation performance during training, averaged across 3 random seeds. The x-axis shows the time steps and the y-axis shows the cumulative returns. Observe that 5-step generalized SIL (blue) generally outperforms other baselines. Tasks with *DM* are from DeepMind Control Suite, and tasks with *(B)* are from Bullet.

We make several observations: (1) For uncorrected $n$-step, the best $n$ is task dependent. However, 5-step generalized SIL consistently improves the performance over uncorrected $n$-step TD3 baselines; (2) SIL losses generally accelerate the optimization. Indeed, both generalized SIL and return-based SIL generally performs better than pure TD3 algorithms; (3) The advantage of generalized SIL is more than $n$-step bootstrap. Because $n$-step generalized SIL is similar to $n$-step updates, it is reasonable to speculate that the performance gains of SIL are partly attributed to $n$-step updates. By the significant advantages of generalized SIL relative to $n$-step updates, we see that its performance gains also come from the lower bound techniques; (4) $n$-step SIL with $n = 5$ works the best. With $n = 1$, SIL does not benefit from bootstrapping partial trajectories with long horizons; with $n = \infty$, SIL does not benefit from bootstrapped values at all. As discussed in Section 3, $n$-step bootstrap provides benefits in (i) variance reduction (replacing the discounted sum of rewards by a value function) and (ii) tightened bounds. In deterministic environment with deterministic policy, the advantage (ii) leads to most of the performance gains.

## 5.2    Ablation study for deterministic actor-critic

Please refer to Table 1 in Appendix D for a summary of ablation experiments over SIL variants. Here, we focus on discussions of the ablation results.

**Horizon parameter** $n$. In our experience, we find that $n = 5$ works reasonably well though other close values might work as well. To clarify the extreme effect of $n$: at one extreme, $n = 1$ and SIL does not benefit from trajectory-based learning and generally underperforms $n = 5$; when $n = \infty$, the return-based SIL does not provide as significant speed up as $n = 5$.

**Prioritized experience replay.** In general, prioritized replay has two hyper-parameters: $\alpha$ for the degree of prioritized sampling and $\beta$ for the degree of corrections [28]. For general SIL, we adopt $\alpha = 0.6, \beta = 0.1$ as in [8]. We also consider variants where the tuples are sampled according to the priority but IS weights are not corrected ($\alpha = 0.6, \beta = 0.0$) and where there is no prioritized sampling ($\alpha = \beta = 0.0$). The results are reported in Table 2 in Appendix D. We observe that generalized SIL works the best when both prioritized sampling and IS corrections are present.

**Over-estimation bias.** Algorithms with over-estimation bias (e.g. DDPG) does not benefit as much (e.g. TD3) from the lower bound loss, as shown by additional results in Appendix D. We speculate that this is because by construction the Q-function network $Q_\theta(x, a)$ should be a close approximation to the Q-function $Q^\pi(x, a)$. In cases where over-estimation bias is severe, this assumption does not hold. As a result, the performance is potentially harmed instead of improved by the *uncontrolled* positive bias [30, 27]. This contrasts with the *controlled* positive bias of SIL, which improves the performance.

### 5.3 Stochastic actor-critic

**Baselines.** For the stochastic actor-critic algorithm, we adopt proximal policy optimization (PPO) [18]. Unlike critic-based algorithms such as TD3, PPO estimates gradients using near on-policy samples.

**Delayed reward environments.** Delayed reward environment tests algorithms' capability to tackle delayed feedback in the form of sparse rewards [8]. In particular, a standard benchmark environment returns dense reward $r_t$ at each step $t$. Consider accumulating the reward over $d$ consecutive steps and return the sum at the end $k$ steps, i.e. $r'_t = 0$ if $t \bmod k \neq 0$ and $r'_t = \sum_{\tau=t-d+1}^{t} r_\tau$ if $t \bmod d = 0$.

**Evaluations.** We compare three baselines: PPO, PPO with SIL [8] and PPO with generalized SIL with $n = 5$-step. We train these variants on a set of OpenAI gym tasks with delayed rewards, where the delays are $k \in \{1, 5, 10, 20\}$. Please refer to Appendix D for further details of the algorithms. The final performance of algorithms after training ($5 \cdot 10^6$ steps for HalfCheetah and $10^7$ for the others) are shown in Figure 3. We make several observations: **(1)** The performance of PPO is generally inferior to its generalized SIL or SIL extensions. This implies the necessity of carrying out SIL in general, as observed in [8]; **(2)** The performance of generalized SIL with $n = 5$ differ depending on the tasks. SIL works significantly better with Ant, while generalized SIL works better with Humanoid. Since SIL is a special case for $n \to \infty$, this implies the potential benefits of adapting $n$ for each task.

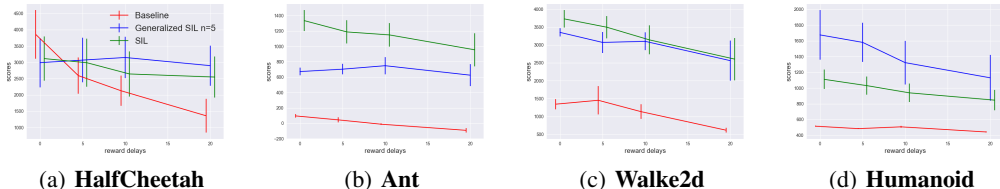

| (a) **HalfCheetah** | (b) **Ant** | (c) **Walke2d** | (d) **Humanoid** |

Figure 3: Standard evaluations on 4 benchmark OpenAI gym tasks. Different colors represent different algorithmic variants. Each curve shows the mean $\pm 0.5$std of evaluation performance at the end of training, averaged across 5 random seeds. The x-axis shows the delayed time steps for rewards and the y-axis shows the cumulative returns. The ticks $\{1, 5, 10, 20\}$ show the delays and the x-axis of the plotted data is slightly shifted for better visualization.

# 6 Further Discussions on Related Work

**Over-estimation bias in Q-learning.** Q-learning and TD-learning are popular algorithms for RL [35, 36]. Due to the max operator, sampled updates of Q-learning naturally incur over-estimation bias, which potentially leads to unstable learning. To mitigate the bias, prior work has considered Double Q-learning[34, 30], explicit bias correction [37], linear combination between Double Q-learning and Q-learning [38], bootstrapping from past predictions [39] and using an ensemble of Q-functions [40]. Similar ideas have been applied to actor-critic algorithms [27]. While it is conventionally believed that over-estimation bias is hurtful to the performance, [40] provides concrete examples where estimation bias (under- or over-estimation) could accelerate learning. In practice, for certain RL environments where rewards are sparse, it is desirable to introduce positive bias to encourage exploration [8].

**Learning from off-policy data.** Off-policy learning is crucial for modern RL algorithms [41, 42]. At the core of many off-policy learning algorithms [1, 43, 44, 3, 45, 46], importance sampling (IS) corrects for the distributional mismatch between behavior $\pi$ and target policy $\mu$, generating *unbiased* updates. Despite the theoretical foundations, IS-based algorithms often underperform empirically motivated algorithms such as $n$-step updates [4–6]. In general, uncorrected $n$-step algorithms could be interpreted as trading-off fast contractions with fixed point bias [7], which seems to have a significant practical effect. In addition to potentially better performance, uncorrected $n$-step updates also do not require e.g. $\mu(a \mid x)$. This entails learning with truly arbitrary off-policy data. Built on top of $n$-step updates, we propose generalized $n$-step SIL which intentionally introduces a positive bias into the fixed point, effectively filtering out behavior data with poor performance. This idea of learning from good-performing off-policy data is rooted in algorithmic paradigms such as behavior cloning [47], inverse RL [48], and more recently instantiated by e.g., episodic control [49, 50] lower bound Q-learning [11] and SIL [8–10].

# 7 Conclusion

We have proposed generalized $n$-step lower bound Q-learning, a strict generalization of return-based lower bound Q-learning and the corresponding self-imitation learning algorithm [8]. We have drawn close connections between $n$-step lower bound Q-learning and uncorrected $n$-step updates: both techniques achieve performance gains by invoking a trade-off between contraction rates and fixed point bias of the evaluation operators. Empirically, we observe that the positive bias induced by lower bound Q-learning provides more consistent improvements than arbitrary $n$-step bias. It is of interest to study in general what bias could be beneficial to policy optimization, and how to exploit such bias in practical RL algorithms.

# 8 Broader Impact

Algorithms which learn from off-policy samples are critical for the applications of RL to more impactful real life domains such as autonomous driving and health care. Our work provides insights into SIL, and its close connections to popular off-policy learning techniques such as $n$-step Q-learning. We believe our work entails a positive step towards better understanding of efficient off-policy RL algorithms, which paves the way for future research into important applications.

# 9 Acknowledgements

The author thanks Mark Rowland and Tadashi Kozuno for insightful discussions about this project.

## Footnotes

[1]By definition, the bias should be the difference between the fixed point $\tilde{Q}$ and target $Q^\pi$. Since TD3 employs heavy replay during training, we expect the Q-function to be close to the fixed point $Q_\theta \approx \tilde{Q}$. Because both the dynamics and policy are deterministic, an one-sample estimate of Q-function is accurate enough to approximate the true Q-function $\hat{Q}^\pi = Q^\pi$. Hence here the bias is approximated by $Q_\theta - \hat{Q}^\pi$.

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
