[Supplementary Material]

# A   Proof of Theorem 1

Recall that under maximum entropy RL, the Q-function is defined as $Q_{\text{ent}}^\pi(x_0, a_0) \coloneqq \mathbb{E}_\pi[r_0 + \sum_{t=1}^\infty \gamma^t(r_t + c\mathcal{H}^\pi(x_t))]$ where $\mathcal{H}^\mu(x_t)$ is the entropy of the distribution $\pi^\mu(\cdot \mid x_t)$. The Bellman equation for Q-function is naturally

$$Q^\pi(x_0, a_0) = \mathbb{E}_\pi[r_0 + \gamma c\mathcal{H}^\pi(x_1) + \gamma Q^\pi(x_1, a_1)].$$

Let the optimal policy be $\pi_{\text{ent}}^*$. The relationship between the optimal policy and its Q-function is $\pi_{\text{ent}}^*(a \mid x) \propto \exp(Q_{\text{ent}}^{\pi_{\text{ent}}^*}(x, a)/c)$. We seek to establish $Q_{\text{ent}}^{\pi_{\text{ent}}^*}(x_0, a_0) \geq \mathbb{E}_\mu[r_0 + \gamma c\mathcal{H}^\mu(x_1) + \sum_{t=1}^{T-1} \gamma^t(r_t + c\mathcal{H}^\mu(x_{t+1})) + \gamma^T Q_{\text{ent}}^\pi(x_T, a_T)]$ for any policy $\mu, \pi$.

We prove the results using induction. For the base case $T = 1$,

$$
\begin{aligned}
Q_{\text{ent}}^{\pi_{\text{ent}}^*}(x_0, a_0) &= \mathbb{E}_{\pi_{\text{ent}}^*}[r_0 + \gamma c\mathcal{H}^{\pi_{\text{ent}}*}(x_1) + \gamma Q_{\text{ent}}^{\pi_{\text{ent}}^*}(x_1, a_1)] \\
&= \mathbb{E}_{x_1 \sim p(\cdot \mid x_0, a_0)}\big[r_0 + \gamma c\mathcal{H}^{\pi_{\text{ent}}*}(x_1) + \gamma \mathbb{E}_{\pi_{\text{ent}}^*}[Q_{\text{ent}}^{\pi_{\text{ent}}^*}(x_1, a_1)]\big] \\
&\geq \mathbb{E}_{x_1 \sim p(\cdot \mid x_0, a_0)}\big[r_0 + \gamma c\mathcal{H}^\mu(x_1) + \gamma \mathbb{E}_\mu[Q_{\text{ent}}^{\pi_{\text{ent}}^*}(x_1, a_1)]\big] \\
&\geq \mathbb{E}_{x_1 \sim p(\cdot \mid x_0, a_0)}\big[r_0 + \gamma c\mathcal{H}^\mu(x_1) + \gamma \mathbb{E}_\mu[Q_{\text{ent}}^\pi(x_1, a_1)]\big].
\end{aligned}
$$
(10)

In the above, to make the derivations clear, we single out the reward $r_0$ and state $x_1 \sim p(\cdot \mid x_0, a_0)$, note that the distributions of these two quantities do not depend on the policy. The first inequality follows from the fact that $\pi_{\text{ent}}^*(\cdot \mid x) = \arg\max_\pi[c\mathcal{H}^\pi(x) + \mathbb{E}_{a \sim \pi(\cdot \mid x)}Q_{\text{ent}}^\pi(x, a)]$. The second inequality follows from $Q_{\text{ent}}^{\pi_{\text{ent}}^*}(x, a) \geq Q_{\text{ent}}^\pi(x, a)$ for any policy $\pi$.

With the base case in place, assume that the result holds for $T \leq k - 1$. Consider the case $T = k$

$$
\begin{aligned}
\mathbb{E}_\mu[r_0 + \gamma c\mathcal{H}^\mu(x_1) &+ \sum_{t=1}^{T-1} \gamma^t(r_t + c\mathcal{H}^\mu(x_{t+1})) + \gamma^T Q_{\text{ent}}^\pi(x_T, a_T)] \\
&\leq \mathbb{E}_\mu\big[r_0 + \gamma c\mathcal{H}^\mu(x_1) + \gamma \mathbb{E}_\mu[Q_{\text{ent}}^{\pi_{\text{ent}}^*}(x_1, a_1)]\big] \\
&\leq Q_{\text{ent}}^{\pi_{\text{ent}}^*}(x_0, a_0),
\end{aligned}
$$

When $\pi = \mu$ we have the special case $\mathbb{E}_\mu[\sum_{t=0}^\infty \gamma^t r_t] \leq V^{\pi^*}(x_0)$, the lower bound which motivated the original lower-bound Q-learning based self-imitation learning [8].

# B   Proof of Theorem 2

For notational simplicity, let $\mathcal{U} \coloneqq (\mathcal{T}^\mu)^{n-1}\mathcal{T}^\pi$ and let $\tilde{\mathcal{U}}Q(x, a) \coloneqq Q(x, a) + [\mathcal{U}Q(x, a) - Q(x, a)]_+$. As a result, we could write $\mathcal{T}_{n,\text{sil}}^{\alpha,\beta} = (1 - \beta)\mathcal{T}^\pi + (1 - \alpha)\beta\tilde{\mathcal{U}} + \alpha\beta\mathcal{U}$.

First, we prove the contraction properties of $\mathcal{T}_{\beta,n,\text{sil}}^\mu$. Note that by construction $|\tilde{\mathcal{U}}Q_1(x, a) - \tilde{\mathcal{U}}Q_2(x, a)| \leq \max(|Q_1(x, a) - Q_2(x, a)|, |\mathcal{U}Q_1(x, a) - \mathcal{U}Q_2(x, a)|) \leq \, \| Q_1 - Q_2 \|_\infty$. Then through the triangle inequality, $\| \mathcal{T}_{n,\text{sil}}^{\alpha,\beta}Q_1 - \mathcal{T}_{n,\text{sil}}^{\alpha,\beta}Q_2 \|_\infty \leq (1 - \beta) \| \mathcal{T}^\pi Q_1 - \mathcal{T}^\pi Q_2 \|_\infty + (1 - \alpha)\beta \| \tilde{\mathcal{U}}Q_1 - \tilde{\mathcal{U}}Q_2 \|_\infty + \alpha\beta \| \mathcal{U}Q_1 - \mathcal{U}Q_2 \|_\infty \leq [(1 - \beta)\gamma + (1 - \alpha)\beta + \alpha\beta\gamma^n] \| Q_1 - Q_2 \|_\infty$. This proves the upper bound on the contraction rates of $\mathcal{T}_{n,\text{sil}}^{\alpha,\beta}$. Let $\eta(\alpha, \beta) = (1 - \beta)\gamma + (1 - \alpha)\beta + \alpha\beta\gamma^n$ and set $\eta(\alpha, \beta) < \gamma$, we deduce $\alpha > \frac{1-\gamma}{1-\gamma^n}$.

Next, we show properties of the fixed point $\tilde{Q}^{\alpha,\beta}$. This point uniquely exists because $\Gamma(\mathcal{T}_{n,\text{sil}}^{\alpha,\beta}) < 1$ if $(1 - \alpha)\beta < 1$. From $\mathcal{T}_{n,\text{sil}}^{\alpha,\beta}\tilde{Q}^{\alpha,\beta} = \tilde{Q}^{\alpha,\beta}$, we could derive by rearranging terms $(1 - \beta)(\mathcal{T}^\pi\tilde{Q} - \tilde{Q}) + \alpha\beta(\mathcal{U}\tilde{Q} - \tilde{Q}) = -(1 - \alpha)\beta(\tilde{\mathcal{U}}\tilde{Q} - \tilde{Q}) \leq 0$. This further implies that $\mathcal{T}^\pi\tilde{Q} \leq \tilde{Q}$. Now let $\mathcal{T} \coloneqq \frac{(1-\beta)}{1-\beta+\alpha\beta}\mathcal{T}^\pi + \frac{\alpha\beta}{1-\beta+\alpha\beta}\mathcal{U}$. This simplifies to $\mathcal{T}\tilde{Q} - \tilde{Q} \leq 0$. By the monotonicity of $\mathcal{T}$, we see $Q^{t\pi+(1-t)\mu^{n-1}\pi} \geq \lim_{k\to}(\mathcal{T})^k\tilde{Q} = Q^\pi$ where $t = \frac{1-\beta}{1-\beta+\alpha\beta}$.

For the another set of inequalities, define $\tilde{H}Q \coloneqq (1 - \beta)\mathcal{T}^* + (1 - \alpha)\beta\tilde{\mathcal{U}}Q + \alpha\beta(\mathcal{T}^*)^n$, where recall that $\mathcal{T}^*$ is the optimality Bellman operator.

First, note $\tilde{H}$ has $Q^{\pi^*}$ as its unique fixed point. To see why, let $\tilde{Q}$ be a generic fixed point of $\tilde{H}$ such that $\tilde{H}\tilde{Q} = \tilde{Q}$. By rearranging terms, it follows that $(1-\beta)(\mathcal{T}^*\tilde{Q} - \tilde{Q}) + \alpha\beta((\mathcal{T}^*)^n\tilde{Q} - \tilde{Q}) = -(1-\alpha)\beta(\tilde{\mathcal{U}}\tilde{Q} - \tilde{Q}) \leq 0$. However, by construction $(\mathcal{T}^*)^i Q \geq Q, \forall i \geq 1, \forall Q$. This implies that $(1-\beta)(\mathcal{T}^*\tilde{Q} - \tilde{Q}) + \alpha\beta((\mathcal{T}^*)^n\tilde{Q} - \tilde{Q}) \geq 0$. As a result, $(1-\beta)(\mathcal{T}^*\tilde{Q} - \tilde{Q}) + \alpha\beta((\mathcal{T}^*)^n\tilde{Q} - \tilde{Q}) = 0$ and $\tilde{Q}$ is a fixed point of $t\mathcal{T}^* + (1-t)(\mathcal{T}^*)^n$. Since $t\mathcal{T}^* + (1-t)(\mathcal{T}^*)^n$ is strictly contractive as $\Gamma(t\mathcal{T}^* + (1-t)(\mathcal{T}^*)^n) \leq t\gamma + (1-t)\gamma^n \leq \gamma < 1$, its fixed point is unique. It is straightforward to deduce that $Q^{\pi^*}$ is a fixed point of $t\mathcal{T}^* + (1-t)(\mathcal{T}^*)^n$ and we conclude that the only possible fixed point of $\tilde{H}$ is $\tilde{Q} = Q^{\pi^*}$. Finally, recall that by construction $\tilde{H}Q \geq Q, \forall Q$. By monotonicity, $Q^{\pi^*} = \lim_{k\to\infty}(\tilde{H})^k \tilde{Q}^{\alpha,\beta} \geq \tilde{Q}^{\alpha,\beta}$. In conclusion, we have shown $Q^{t\pi + (1-t)\mu^{n-1}\mu} \leq \tilde{Q}^{\alpha,\beta} \leq Q^{\pi^*}$.

## C  Additional theoretical results

**Theorem 3.** *Let $\pi^*$ be the optimal policy and $V^{\pi^*}$ its value function under standard* RL *formulation. Given a partial trajectory $(x_t, a_t)_{t=0}^n$, the following inequality holds for any $n$,*

$$V^{\pi^*}(x_0) \geq \mathbb{E}_\mu\Big[\sum_{t=0}^{n-1} \gamma^t r_t + \gamma^n V^\pi(x_k)\Big] \tag{11}$$

*Proof.* Let $\pi, \mu$ be any policy and $\pi^*$ the optimal policy. We seek to show $V^{\pi^*}(x_0) \geq \mathbb{E}_\mu[\sum_{t=0}^{T-1} \gamma^t r_t + \gamma^T V^\pi(x_T)]$ for any $T \geq 1$.

We prove the results using induction. For the base case $T = 1$, $V^{\pi^*}(x_0) = \mathbb{E}_{\pi^*}[Q^{\pi^*}(x_0, a_0)] \geq \mathbb{E}_\mu[Q^{\pi^*}(x_0, a_0)] = \mathbb{E}_\mu[r_0 + \gamma V^{\pi^*}(x_1)] \geq \mathbb{E}_\mu[r_0 + \gamma V^\pi(x_1)]$, where the first inequality comes from the fact that $\pi^*(\cdot \mid x_0) = \arg\max_a Q^{\pi^*}(x_0, a)$. Now assume that the statement holds for any $T \leq k - 1$, we proceed to the case $T = k$.

$$\mathbb{E}_\mu\Big[\sum_{t=0}^{k-1} \gamma^t r_t + \gamma^k V^\pi(x_k)\Big] = \mathbb{E}_\mu\Big[r_0 + \gamma\mathbb{E}_\mu\big[\sum_{t=0}^{k-2} \gamma^t r_t + \gamma^{k-1} V^\pi(x_k)\big]\Big]$$
$$\leq \mathbb{E}_\mu[r_0 + \gamma V^{\pi^*}(x_1)] \leq V^{\pi^*}(x_0),$$

where the first inequality comes from the induction hypothesis and the second inequality follows naturally from the base case. This implies that $n$-step quantities of the form $V^{\pi^*}(x_0) \geq \mathbb{E}_\mu[\sum_{t=0}^{n-1} \gamma^t r_t + \gamma^n V^\pi(x_T)]$ are lower bounds of the optimal value function $V^{\pi^*}(x_0)$ for any $n \geq 1$. $\qquad\square$

## D  Experiment details

**Implementation details.** The algorithmic baselines for deterministic actor-critic ( TD3 and DDPG) are based on OpenAI Spinning Up https://github.com/openai/spinningup [51]. The baselines for stochastic actor-critic is based on PPO [18] and SIL+PPO [8] are based on the author code base https://github.com/junhyukoh/self-imitation-learning. Throughout the experiments, all optimizations are carried out via Adam optimizer [52].

**Architecture.** Deterministic actor-critic baselines, including TD3 and DDPG share the same network architecture following [51]. The Q-function network $Q_\theta(x, a)$ and policy $\pi_\phi(x)$ are both 2-layer neural network with $h = 300$ hidden units per layer, before the output layer. Hidden layers are interleaved with $\mathrm{relu}(x)$ activation functions. For the policy $\pi_\phi(x)$, the output is stacked with a $\tanh(x)$ function to ensure that the output action is in $[-1, 1]$. All baselines are run with default hyper-parameters from the code base.

Stochastic actor-critic baselines (e.g. PPO) implement value function $V_\theta(x)$ and policy $\pi_\phi(a \mid x)$ both as 2-layer neural network with $h = 64$ hidden units per layer and tanh activation. The stochastic policy $\pi_\phi(a \mid x)$ is a Gaussian $a \sim \mathcal{N}(\mu_\phi(x), \sigma^2)$ with state-dependent mean $\mu_\phi(x)$ and a global variance parameter $\sigma^2$. Other missing hyper-parameters take default values from the code base.

Table 1: Summary of the performance of algorithmic variants across benchmark tasks. We use *uncorrected* to denote prioritized sampling without IS corrections. Return-based SIL is represented as SIL with $n = \infty$. For each task, algorithmic variants with top performance are highlighted (two are highlighted if they are not statistically significantly different). Each entry shows mean $\pm$ std performance.

| Tasks | SIL $n = 5$ | SIL $n = 5$ (uncorrected) | SIL $n = 1$ (uncorrected) | 5-step | 1-step | SIL $n = \infty$ |
|---|---|---|---|---|---|---|
| DMWALKERRUN | **642 $\pm$ 107** | **675 $\pm$ 15** | 500 $\pm$ 138 | 246 $\pm$ 49 | 274 $\pm$ 100 | 320 $\pm$ 111 |
| DMWALKERSTAND | **979 $\pm$ 2** | 947 $\pm$ 18 | 899 $\pm$ 55 | 749 $\pm$ 150 | 487 $\pm$ 177 | 748 $\pm$ 143 |
| DMWALKERWALK | 731 $\pm$ 151 | 622 $\pm$ 197 | 601 $\pm$ 108 | **925 $\pm$ 10** | 793 $\pm$ 121 | 398 $\pm$ 203 |
| DMCHEETAHRUN | **830 $\pm$ 36** | 597 $\pm$ 64 | 702 $\pm$ 72 | 553 $\pm$ 92 | 643 $\pm$ 83 | 655 $\pm$ 59 |
| ANT | **4123 $\pm$ 364** | 3059 $\pm$ 360 | **3166 $\pm$ 390** | 1058 $\pm$ 281 | 3968 $\pm$ 401 | 3787 $\pm$ 411 |
| HALFCHEETAH | 8246 $\pm$ 784 | 9976 $\pm$ 252 | **10417 $\pm$ 364** | 6178 $\pm$ 151 | **10100 $\pm$ 481** | 8389 $\pm$ 386 |
| ANT(B) | **2954 $\pm$ 54** | 1690 $\pm$ 564 | 1851 $\pm$ 416 | **2920 $\pm$ 84** | 1866 $\pm$ 623 | 1884 $\pm$ 631 |
| HALFCHEETAH(B) | **2619 $\pm$ 129** | **2521 $\pm$ 128** | 2420 $\pm$ 109 | 1454 $\pm$ 338 | 2544 $\pm$ 31 | 2014 $\pm$ 378 |

## D.1 Further implementation and hyper-parameter details

**Generalized SIL for deterministic actor-critic.** We adopt TD3 [27] as the baseline for deterministic actor-critic. TD3 maintains a Q-function network $Q_\theta(x, a)$ and a deterministic policy network $\pi_\theta(x)$ with parameter $\theta$. The SIL subroutines adopt a prioritized experience replay buffer: the return-based SIL samples tuples according to the priority $[R^\mu(x, a) - Q_\theta(x, a)]_+$ and minimizes the loss function $[R^\mu(x, a) - Q_\theta(x, a)]_+$; the generalized SIL samples tuples according to the priority $[L^{\pi, \mu, n}(x, a) - Q_\theta(x, a)]_+$ and minimizes the loss function $[L^{\pi, \mu, n}(x, a) - Q_\theta(x, a)]_+$. The experience replay adopts the parameter $\alpha = 0.6, \beta = 0.1$ [53]. Throughout the experiments, TD3-based algorithms all employ $\alpha = 10^{-3}$ for the network updates.

To calculate the update target $L^{\pi, \mu, n}(x_0, a_0) = \sum_{t=0}^{n-1} \gamma^t r_t + Q_{\theta'}(x_n, \pi_{\theta'}(x_n))$ with partial trajectory $(x_t, a_t, r_t)_{t=0}^n$ along with the target value network $Q_{\theta'}(x, a)$ and policy network $\pi_{\theta'}(x)$. The target network is slowly updated as $\theta' = \tau\theta' + (1 - \tau)\theta$ where $\tau = 0.995$ [14].

**Generalized SIL for stochastic actor-critic.** We adopt PPO [18] as the baseline algorithm and implement modifications on top of the SIL author code base https://github.com/junhyukoh/self-imitation-learning as well as the original baseline code https://github.com/openai/baselines [54]. All PPO variants use the default learning rate $\alpha = 3 \cdot 10^{-4}$ for both actor $\pi_\theta(a \mid x)$ and critic $V_\theta(x)$. The SIL subroutines are implemented as a prioritized replay with $\alpha = 0.6, \beta = 0.1$. For other details of SIL in PPO, please refer to the SIL paper [8].

The only difference between generalized SIL and SIL lies in the implementation of the prioritized replay. SIL samples tuples according to the priority $[R^\mu(x, a) - V_\theta(x)]_+$ and minimize the SIL loss function $([R^\mu(x, a) - V_\theta(x)]_+)^2$ for the value function, and $-\log \pi_\theta(a \mid x)[R^\mu(x, a) - V_\theta(x)]_+$ for the policy. Generalized SIL samples tuples according to the priority $([L^{\pi, \mu, n}(x, a) - V_\theta(x)]_+)^2$, and minimize the loss $([L^{\pi, \mu, n}(x, a) - V_\theta(x)]_+)^2$ and $-\log \pi_\theta(a \mid x)[L^{\pi, \mu, n}(X, a) - V_\theta(x)]_+$ for the value function/policy respectively.

To calculate the update target $L^{\pi, \mu, n}(x_0, a_0) = \sum_{t=0}^{n-1} \gamma^t r_t + V_{\theta'}(x_n)$ with partial trajectory $(x_t, a_t, r_t)_{t=0}^n$ along with the target value network $V_{\theta'}(x)$. We apply the target network technique to stabilizie the update, where $\theta'$ is a delayed version of the major network $\theta$ and is updated as $\theta' = \tau\theta' + (1 - \tau)\theta$ where $\tau = 0.995$.

## D.2 Additional experiment results

**Comparison across related baselines.** We make clear the comparison between related baselines in Table 1. We present results for $n$-step TD3 with $n \in \{1, 5\}$; TD3 with generalized SIL with $n = 5$ and its variants with different setups for prioritized sampling; TD3 with return-based SIL ($n = \infty$). We show the results across all 8 tasks - in each entry of Table 1 we show the mean $\pm$ std of performance averaged over 3 seeds. The performance of each algorithmic variant is the average testing performance of the last $10^4$ training steps (from a total of $10^6$ training steps). The best

Table 2: Comparison between different replay schemes. For each task, algorithmic variants with top performance are highlighted (two are highlighted if they are not statistically significantly different). Each entry shows mean ± std performance.

| Tasks | SIL $n = 5$ | SIL $n = 5$ (uncorrected) | SIL $n = 5$ (no priority) |
|---|---|---|---|
| DMWalkerRun | **642 ± 107** | **675 ± 15** | 424 ± 127 |
| DMWalkerStand | **979 ± 2** | 947 ± 18 | 634 ± 184 |
| DMWalkerWalk | **731 ± 151** | 622 ± 197 | **766 ± 103** |
| DMCheetahRun | **830 ± 36** | 597 ± 64 | **505 ± 182** |
| Ant | **4123 ± 364** | 3059 ± 360 | 4358 ± 496 |
| HalfCheetah | 8246 ± 784 | **9976 ± 252** | 8927 ± 596 |
| Ant(B) | **2954 ± 54** | 1690 ± 564 | **2910 ± 88** |
| HalfCheetah(B) | **2619 ± 129** | **2521 ± 128** | 2284 ± 85 |

(a) **DMWalkerRun**  (b) **DMWalkerStand**  (c) **Ant**  (d) **HalfCheetah**

Figure 4: Standard evaluations on 4 simulation tasks for DDPG baselines. Different colors represent different algorithmic variants. Each curve shows the mean ± 0.5std of evaluation performance during training, averaged across 3 random seeds. The x-axis shows the time steps and the y-axis shows the cumulative returns.

algorithmic variant is highlighted in bold. We see that in general generalized SIL with $n = 5$ performs the best.

**Ablation on the prioritized sampling.** In prioritized sampling [53], when the tuples $d = (x_i, a_i, r_i)_{i=0}^n \in \mathcal{D}$ are sampled with priorities $s_d$, it is sampled with probability $p(d) \propto s_d^\alpha$. During updates, the IS correction consists in optimizing the loss $\mathbb{E}_d[w_d l_d]$ where $l_d$ is the loss computed from tuple $d$ and the IS correction weight $w_d = (N \cdot p_d)^{-\beta}$ where $N$ is the number of tuples in the buffer $\mathcal{D}$.

We compare several prioritized sampling variants of generalized SIL in Table 2. There are three variants: SIL $n = 5$ with both prioritized sampling ($\alpha = 0.6$) and IS correction ($\beta = 0.1$); SIL $n = 5$ with prioritized sampling ($\alpha = 0.6$) only and without IS correction ($\beta = 0.0$); SIL $n = 5$ with no prioritized sampling ($\alpha = \beta = 0.0$). The performance setup in Table 2 is the same as in Table 1. It can be seen from Table 2 that generalized SIL performs the best with full prioritized sampling.

**Results on DDPG.** DDPG is a baseline actor-critic algorithm with a deterministic actor [15]. Compared to TD3, DDPG does not adopt a double-critic approach [27] and suffers from over-estimation bias of the Q-function [30].

We present the baseline evaluation result of DDPG in Figure 4, where we show the results for a few variants: DDPG with $n$-step update, $n \in \{1, 5\}$; DDPG with generalized SIL $n = 5$ and DDPG with return-based SIL ($n = \infty$). We see that the performance gains of DDPG with generalized SIL $n = 5$ are not as significant - indeed, overall DDPG with $n = 5$ has the best performance. We speculate that this is partly due to the over-estimation bias of DDPG: the formulation of generalized SIL is motivated by shifting the fixed point $Q^\pi$ with an positive bias. The baseline algorithm benefits the most from generalized SIL when indeed in practice $Q_\theta \approx Q^\pi$. However, this is not the case for DDPG as the algorithm already has high positive bias in that $Q^\theta > Q^\pi$, which reduces the potential gains that come from generalized SIL.