[Reviews · NeurIPS 2020]

Review 1

Summary and Contributions: The paper proposes an extension to self-imitation learning, generalising return-based lower-bound Q-learning to a family of n-step lower bound Q-learning. It is shown that lower-bound Q-learning trades off fixed point bias and contraction rate, compared to uncorrected n-step Q-learning. Empirical evidence shows improvements in RL benchmarks.

Strengths: The proposed method generalises return-based lower-bound Q-learning to n-step settings, introducing a flexible family of new self-imitation learning algorithms. The change is intuitive as n-step version leverages the current Q-function and reduces variance. In addition, the paper shows the trade-off between contraction rate and fixed-point bias between lower-bound Q-learning and uncorrected n-step.

Weaknesses: I have several concerns about the proposed method: 1) It is not quite clear how the trade-off between contraction rate and fixed point bias affects practical performances. Additional discussion over this could strengthen the relationship between the theoretical and the practical results. 2) The appears to be a disconnect between empirical evaluation and the theoretical discussion. In general, the best performing models from the experiments use importance-sampling (IS), which is not reflected in the theoretical results. More discussion about this aspect is needed. 3) It would be beneficial to include empirical comparison with unbiased Q-function estimator, such as Retrace and \alpha-Retrace to understand the trade-off of the introduced bias.

Correctness: The method is theoretically justified and the proof appears correct. The ablation study on the proposed n-step Q-function estimator is informative and quite comprehensive.

Clarity: The paper is well written, and easy to follow.

Relation to Prior Work: There is extensive discussion with respect to previous work, and how the proposed method relates to them. Sufficient details are given about how the proposed method differs from previous works.

Reproducibility: Yes

Additional Feedback: -- After rebuttal My concerns are adequately addressed. I hope that the authors incorporate their revised discussion about contraction rate and fixed point bias, with respect to empirical performances. I revised my score to a 7.


Review 2

Summary and Contributions: Self-imitation learning (SIL) has shown good performance in some hard exploration tasks. This paper proposes the generalised self-imitation learning (SIL), which generalises the SIL method with n-step lower bound Q-learning. The n-step lower bound Q-learning provides a trade-off between fixed point bias and contraction rate, which connects to the uncorrected n-step Q-learning. In the experiments, the generalised method shows better performance in some testing environments. However, the improvements are not very strong.

Strengths: The paper has a good theoretical grounding, which derives the n-step generalised Self-Imitation Learning via the maximum-entropy RL formulation. The contribution of the paper includes a generalised self-imitation learning method, which proves additional advantages over the original SIL method. It learns from partial trajectories. The authors also formalise the trade-offs of SIL and show that the generalised SIL trades-off contraction rates with fixed point bias. The authors show that during evaluation, the proposed method, generalised SIL outperform the baseline methods.

Weaknesses: The performance improvement is incremental and needs to be further evaluated. For example, each experiment should be conducted over 5 random seeds, instead of 3 seeds, for a more accurate comparison. Besides, in only 3 out of 8 environments, shown in Figure 2, the proposed method shows clear improvement. And more baseline methods should be considered, such as SAC. Also, in the original SIL paper, the method also show strong performance in the hard exploration environments, such as Montezuma’s Revenge. So, how does the generalise SIL compare to SIL in the Montezuma’s Revenge task? Does it improve performance or not? Post-rebuttal: After reading the rebuttal, I increase my score because that the authors add some new experimental results comparing with SAC. It would be more convincing if there are results in Montezuma’s Revenge.

Correctness: Yes, these are correct.

Clarity: Yes, it is well written.

Relation to Prior Work: Yes, it is clear.

Reproducibility: Yes

Additional Feedback:


Review 3

Summary and Contributions: The paper proposes a general version of self-imitation learning by replace the original lower bound Q by a n-step generalized lower bound of Q^*, which is tunable and can tradeoff the contraction rate and bias to achieve a better performance. Detailed theoretical discussion and extensive empirical results demonstrate the improvement of the algorithm.

Strengths: The paper is well motivated and the idea is simple to understand. It combines two important aspect of off policy learning methods that can achieve a good performance. The idea of generalizing the lower bound of Q-function in self-imitation learning opens a potential way to improve the performance. The empirical results are convincing and show gain on performance.

Weaknesses: My major concern is for the variance of empirical operator. Though the paper mentioned in line 87-88 that variance in practice could be reduced by large training batch sizes. However, in continuous settings, each (x,a) pair may only has one observed trajectory, where the lower bound L^{\pi, \mu, n} can only be estimated by a single trajectory, which will unavoidable have a large variance. Since the tradeoff lemma in equation (1) includes variance term, I think it is better to address that problem in the paper, e.g. consider an oracle transition operator that can draw more trajectories starting from (x,a), will the performance get improved empirically?

Correctness: The paper is theoretically sound and the empirical methodology is standard.

Clarity: The paper is well written and well motivated. Minor: a few typo: line 78: variance of \T includes an undefine Q, which should be \tilde{Q}? line 108: \mathcal H should be \mathcal H^{\pi};

Relation to Prior Work: Existing works have been clearly discussed in the introduction and throughout the paper. Another line of recent research of off policy learning which involve learning importance sampling over state or state/action pair is also worth mentioned[1-5] which avoid the high variance of product of \pi(a_t|x_t)/\mu(a_t|x_t). 1. Liu, Li, Tang, Zhou: Breaking the Curse of Horizon: Infinite-Horizon Off-Policy Estimation 2. Liu, Swaminathan, Agarwal, Brunskill: Off-Policy Policy Gradient with State Distribution Correction 3. Nachum, Dai, Chow, Li: DualDICE: Behavior-Agnostic Estimation of Discounted Stationary Distribution Corrections 4. Nachum, Dai, Kostrikov, Chow, Li, Schuurmans: AlgaeDICE: Policy Gradient from Arbitrary Experience 5.Jiang, Huang: Minimax Confidence Interval for Off-Policy Evaluation and Policy Optimization.

Reproducibility: Yes

Additional Feedback:

[Author Response · NeurIPS 2020]

We thank the reviewers for their careful and valuable feedback. We address their main points in our comments below.

**[R1] How the trade-off affects practical performance.** This is a good point; we appreciate you raising it. Our paper does *not* seek to make a strong theoretical statement that trading off contraction & bias definitively leads to performance gains. Instead, we focus on identifying the potential operator trade-off with SIL and proposing a generalized $n$-step alternative which opens doors to further exploit the trade-offs. Though empirical evidence suggests that fast contraction tends to practical gains, there is little theoretical explanations, even with [Rowland et al, 2019] which motivates our work. We speculate that the theory is not straightforward because both us and [Rowland et al, 2019] focus on policy evaluation, while the the full algorithm interleaves with optimization, which greatly complicates the analysis. Therefore, we leave a more comprehensive study for future work. Empirically, we find in Table 1 (App. D) that $n$-step SIL with $n = 5$ outperforms $n = 1$. This is consistent with results from prior work that fast contraction tends to empirical gains.

**[R1] Disconnect between theory & experiments.** As an ideal operator, importance sampling (IS) achieves the fastest possible contraction (i.e. it contracts to the fixed point with one iteration) and is unbiased. However, its stochastic estimate has high variance and is rarely used in practice (see [Munos et al, 2016]). As a result, we believe that IS is not the best performing model in practical experiments. Consistent with this argument, most prior work also consider the high variance a major downside of IS [Munos et al, 2016; Rowland et al, 2019]. We will discuss more in the revisions.

**[R1] Comparison to unbiased methods.** For the continuous control tasks that we consider, the baseline algorithm TD3 adopts a deterministic policy and it is not feasible to apply Retrace (which requires stochastic policy to perform truncated IS). As an alternative unbiased baseline, the uncorrected $n = 1$ step generally underperforms $n = 5$-step SIL.

**[R2] Performance.** Thanks for raising this issue. From Fig 2, though SIL with $n = 5$ does not outperform *all* the baseline alternatives on *all* tasks, it consistently ranks as top two among the majority of tasks, indicating its more stable performance. For fair evaluations, we believe it is not reasonable to require $n$-step SIL to outperform the *best* among all other alternative baselines on every task. Instead, we believe it is more reasonable to compare $n$-step SIL with alternative baselines on a one-to-one basis – by such a metric, the improvement is clear. For example, $n$-step SIL clearly outperforms $n$-step uncorrected on 6/8 tasks while outperforms vanilla SIL on 7/8 tasks.

**[R2] Random seeds & SAC.** We agree that running more seeds potentially leads to more accurate assessments. However, we highlight that despite a relatively small number of seeds, in Fig 2 most curves are well separated, indicating statistically significant differences. Note also that the highly cited PPO paper uses 3 seeds across all experiments. Regarding SAC: We did not include SAC baseline for a few reasons: (1) Though we propose a maxent lower bound in Thm 1, all theories on the trade-offs of operators are exclusively derived in the conventional RL setup (including results from [Rowland et al, 2019]). As a result, we do not think comparing to SAC would offer much insights as to echo the theory; (2) We speculate that applying the $n$-step technique in Thm 1 to SAC might not lead to significant gains out of the box, as it might be sensitive to the entropy coeff. In fact, [Oh et al, 2018] derives the SIL formulation under maxent RL, but the entropy term is dropped when calculating the lower bounds in their implementation. In Fig 1, we provide SAC results, which mostly underperform $n$-step SIL, especially on DM suite. We speculate this is because SAC hyperparams have been commonly tuned on gym envs. This corroborates our speculation that SAC performance might be sensitive to the entropy coeff.

| Tasks | SAC |
|---|---|
| DMWALKERRUN | $23 \pm 1$ |
| DMWALKERWALK | $87 \pm 83$ |
| DMWALKERSTAND | $440 \pm 87$ |
| DMCHEETAH | $3 \pm 1$ |
| ANT | $2645 \pm 1462$ |
| HALFCHEETAH | $11451 \pm 406$ |
| ANT(B) | $808 \pm 29$ |
| HALFCHEETAH(B) | $914 \pm 251$ |

Figure 1: The $n$-step SIL outperforms vanilla SAC on most tasks.

**[R2] Montezuma.** We did not include Montezuma as we initially could not replicate the results of [Oh et al, 2018]. We speculate that with proper tuning, $n$-step SIL should outperform typical baselines but might slightly underperform return-based SIL. This is partly because when rewards are sparse, using returns as lower bounds might be more accurate than using learned bootstrapped values. As a result, return-based SIL [Oh et al, 2018] might still be more suitable for tasks with highly sparse rewards as in Montezuma. However, we believe this does not undermine results in this paper, where we highlight the gains of $n$-step SIL on tasks with dense rewards & midly sparse rewards (delayed rewards).

**[R3] Variance of the estimator & related work.** This is a good point, we will discuss more details in the revisions. There a few reasons why the variance is not explicitly addressed: (1) Uncorrected $n$-step & SIL remove all IS ratios, which arguably greatly reduces the variance compared to IS based methods, e.g., Retrace. This is in line with arguments made in prior work such as [Rowland et al, 2019] where the variance is not addressed explicitly; (2) Though from each $(x, a)$ pair there is only one trajectory, the bootstrapped values at the end of the $n$-step are learned and could interpolate between different pairs, which leads to more accurate estimates; (3) Particular to the continuous control tasks where both dynamics and policy are deterministic, one-sample estimate could have relatively low variance. See more discussions at line 263-275. Regarding related work: we are aware of the duality & state marginal method to off-policy evaluation. We will include them as related work and leave their combinations with SIL as future work.

**[ALL REVIEWERS]** We appreciate the other points you have raised that we cannot address in this one-page response; they improve our manuscript and we will adjust our text based on your comments.

[Meta-Review · NeurIPS 2020]

The author response provided satisfactory answers to the concerns of the reviewers with respect to contraction/bias tradeoff, disconnect between the experimental results and theory, and variance of the estimator. This lead one reviewer to increase their score for this paper, which already had reasonably solid scores.